

# Achieving optimal trade-off for student dropout prediction with multi-objective reinforcement learning

Feng Pan[1,2], Hanfei Zhang[1], Xuebao Li[2], Moyu Zhang[1] and Yang Ji[1]

[1] School of Information and Communication Engineering, Beijing University of Posts and Telecommunications, Beijing, China
[2] School of Information Science and Technology, Baotou Teachers' College, Baotou, Inner Mongolia, China

## ABSTRACT

Student dropout prediction (SDP) in educational research has gained prominence for its role in analyzing student learning behaviors through time series models. Traditional methods often focus singularly on either prediction accuracy or earliness, leading to sub-optimal interventions for at-risk students. This issue underlines the necessity for methods that effectively manage the trade-off between accuracy and earliness. Recognizing the limitations of existing methods, this study introduces a novel approach leveraging multi-objective reinforcement learning (MORL) to optimize the trade-off between prediction accuracy and earliness in SDP tasks. By framing SDP as a partial sequence classification problem, we model it through a multiple-objective Markov decision process (MOMDP), incorporating a vectorized reward function that maintains the distinctiveness of each objective, thereby preventing information loss and enabling more nuanced optimization strategies. Furthermore, we introduce an advanced envelope Q-learning technique to foster a comprehensive exploration of the solution space, aiming to identify Pareto-optimal strategies that accommodate a broader spectrum of preferences. The efficacy of our model has been rigorously validated through comprehensive evaluations on real-world MOOC datasets. These evaluations have demonstrated our model's superiority, outperforming existing methods in achieving optimal trade-off between accuracy and earliness, thus marking a significant advancement in the field of SDP.

## INTRODUCTION

Student dropout prediction (SDP) is gaining significant attention as a critical area of research and is commonly modeled using time series classification models (*Prenkaj et al., 2020*; *Janelli & Lipnevich, 2021*; *Xing, 2018*; *Psathas, Chatzidaki & Demetriadis, 2023*). The primary focus of the SDP research community has been on developing algorithms for improving prediction accuracy (*Feng, Tang & Liu, 2019*; *Pulikottil & Gupta, 2020*; *Pan et al., 2022*; *He et al., 2015*). However, this emphasis on accuracy often disregards other key metrics, with recent developments highlighting the significance of prediction

Corresponding author
Yang Ji, jiyang@bupt.edu.cn

timeliness. Early predictions are crucial for identifying students' initial poor learning trends, enabling timely interventions by educators (*Taylor, Veeramachaneni & O'Reilly, 2014*; *Gray & Perkins, 2019*; *Berens et al., 2019*; *He et al., 2015*; *Panagiotakopoulos et al., 2021*). Yet, existing methods typically focused on optimizing a single objective–either accuracy or earliness–resulting in sub-optimal outcomes due to the neglect of potential trade-offs between multiple objectives (*Shi et al., 2020*). Nonetheless, advising a learning policy that considers both accuracy and earliness presents a challenge. On one hand, improving prediction accuracy typically necessitates gathering extensive data, which can compromise the timeliness of prediction, potentially missing early intervention opportunities. On the other hand, ensuring prompt predictions often involves making early-stage predictions with limited data, which can diminish accuracy and make the prediction less reliable for stakeholders (*Zhu et al., 2021*). The inverse relationship between prediction accuracy and earliness turns their simultaneous optimization into a complex, multi-objective dilemma (*Akasiadis et al., 2022*; *Achenchabe et al., 2021*; *Swacha & Muszyńska, 2023*).

In recent years there has been an growing interest in exploring methods for balancing prediction accuracy and earliness in SDP tasks. The literature reveals two main methodologies: post-processing after prediction and in-prediction processing. The former involves training with complete time series data and testing on partial sequences at various preset halting points, and then utilizing metrics like the harmonic mean to determine between accuracy and earliness (*Ben Soussia et al., 2022*; *Dachraoui, Bondu & Cornuéjols, 2015*). These methods, however, suffer from high computational demands and a lack of adaptability due to reliance on fixed halting points and predefined thresholds. The latter, more sophisticated approach, involves real-time adjustment within the model's inference process, either by integrating regularization for balancing objectives (*Deho et al., 2022*) or employing multi-objective optimization (MOO) to manage the trade-offs (*Jimenez et al., 2019*). Despite the promise of these methodologies, the non-convexity of the solution set poses challenges in navigating and pinpointing the global optimum (*Wang et al., 2023*), which complicates effective solution identification. This insight underscores the need for innovative methods that dynamically balance accuracy and earliness, ensuring adaptability and practical efficiency.

Furthermore, the challenge of harmonizing accuracy and earliness extends beyond localized concerns, emphasizing its global significance in improving preemptive interventions across various fields. To address this, some researchers have explored the use of reinforcement learning (RL) to optimize conflicting objectives in the field of early classification of time series, given its ability to make decisions from a long-term perspective (*Martinez et al., 2018*; *Martinez et al., 2020*; *Hartvigsen et al., 2019*). This approach involves training an early classifier using partial sequence data, treating these sequences as environment states, and utilizing an RL agent to choose between immediate classification and waiting for more information. Commonly, the predictive objectives are encoded into several reward functions (*Martinez et al., 2018*; *Martinez et al., 2020*), and a user controlled hyper-parameter is employed to fine tune the proportion of different objectives. A slightly different approach adopts a single scalar reward (*Hartvigsen et al., 2019*) to represent the objective of prediction accuracy, and introduces an additional

loss term with a trade-off hyperparameter to encourage early stop. Through real-time interaction with the environment, these methods enable the agent to adaptively refine its strategy and value functions to maximize rewards. By leveraging RL, models can more efficiently explore the solution space in dynamic environments and develop strategies for the early classification of time series.

The essence of the RL based models lies in using a scalarization function to compress multiple objectives into a singular one. While this approach can simplify the optimization process to some extent, it also has significant drawbacks. The main issue is the potential for losing crucial information about the individual contributions of each objective. Moreover, it may obscure the distinct aspects and interrelations between objectives, potentially leading to a less sophisticated management and understanding of the various goals. This reduction may result in suboptimal decision (*Zhang, Qi & Shi, 2023*), especially in scenarios where objectives conflict and necessitate a careful balance. Additionally, RL agents tend to search for optimal solutions along predefined scalarized directions set by the weight configurations in the scalarization function (*Xu et al., 2021*). This process, depicted as projecting rays from the origin across the objective space in Fig. 1, each ray symbolizing a distinct combination of weights. However, this method may overlook other potential optimal solutions that do not align with these pre-defined directions, leading to local rather than global Pareto optimality (*Xu et al., 2021*). This limitation highlights the need for more sophisticated methodologies that can effectively handle the complexity of multiple objectives, ensuring a broader exploration of the solution space to identify truly optimal strategies.

To address the challenges, we propose a novel approach based on multi-objective reinforcement learning (MORL). We aim to answer the following research questions, **RQ1**: How can reinforcement learning models be designed to effectively capture and optimize the trade-off between accuracy and earliness in predicting student dropout, reflecting the multifaceted objectives inherent in educational settings? **RQ2**: What strategies can be employed in reinforcement learning frameworks to ensure a comprehensive exploration of the solution space, thereby facilitating the identification and implementation of globally optimal strategies for managing the accuracy-earliness trade-off in student dropout prediction? We tried to resolve these questions by modeling the trade-off between accuracy and earliness in SDP as a partial sequence classification task, with the assumption that we do not have access to the full state information but could predict student dropout by incomplete sequence. We formulate this task as a multiple-objective Markov decision process (MOMDP), characterized by a comprehensive set of states, actions, and importantly, a vectorized reward function tailored for each objective. This vector reward directly tackles the issue of information loss by preserving the distinctiveness of each objective's contribution, allowing for a more detailed optimization process. Furthermore, to tackle the problem of optimization towards predefined directions, we incorporate an advanced technique known as envelope Q-learning (*Yang, Sun & Narasimhan, 2019*). This method plays a key role in policy updates, aimed at embracing a broad spectrum of preferences and thus, promoting a comprehensive exploration of the solution space. The practical implementation of this theory is executed through training an early classifier agent using the Multi-Objective Double Deep Q-Network (MODDQN) algorithm. Upon

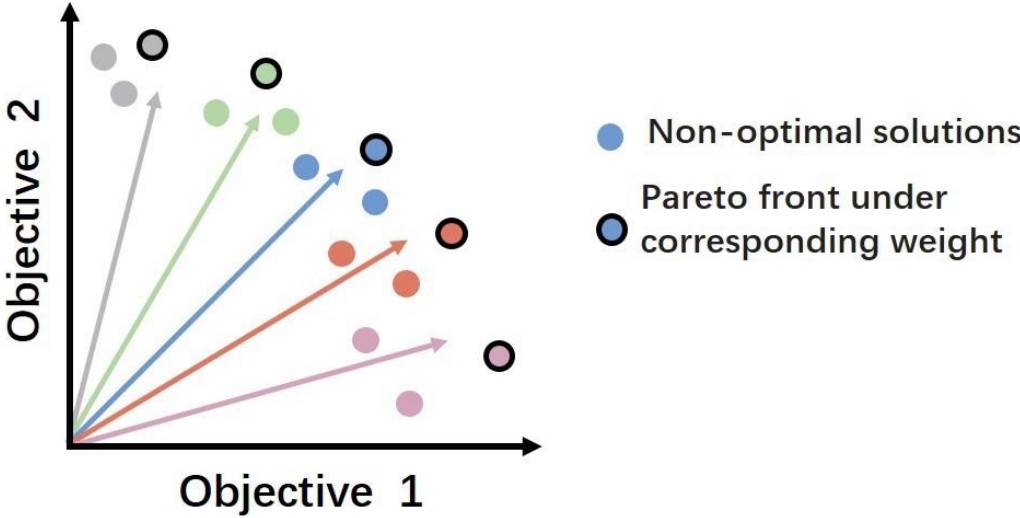

**Figure 1** **An example of a Pareto front that trades off between two different objectives.** Pairs of weights create rays extending radially from the origin, with each circle symbolizing a potential outcome identified during a single objective optimization, as defined by the ray's corresponding color. Circles encased in black borders represent the optimal solutions for their respective weights, collectively constituting a convex approximation of the Pareto front.

completion of the training phase, the selection of an operational strategy can be dynamically achieved through the modification of various objective weights, providing a tailored balance between classification accuracy and earliness. Through these innovations, we empower the early classifier agent with the capability to dynamically select the most appropriate operational strategy by adjusting the weights attributed to various objectives, effectively overcoming the limitations previously associated with scalarization RL methods. Our experimental findings validate the efficacy of the MORL algorithm, especially its proficiency in enabling a flexible trade-off and in identifying Pareto-optimal solutions. Aimed at maximizing student dropout prediction accuracy as promptly as possible, our approach highlights the significant potential of Artificial Intelligence (AI) to revolutionize educational analytics and intervention strategies, thus illustrating a step forward in the application of MORL to complex, multi-objective problems.

The contributions of this article are as follow:

- We design multi-perspective sources of vector reward to train the agent to prevent information loss caused by the dimensionality reduction of summation the multiple rewards into one scala reward.
- We leverage vectorized value functions and perform envelope value updates to train a unified policy network optimized across the full spectrum of preferences within a domain. Consequently, this trained network is capable of generating the optimal policy for any preference specified by the user.
- We evaluate our proposed model using two real-world MOOC datasets. To the best of our knowledge, this is the first work that the MORL method being used in the

accuracy-earliness trade-off issue of SDP. Results show that our method significantly outperforms state-of-the-art approaches in achieving optimal trade-off. This innovative AI application not only advances the field of SDP but also signifies a major leap in the application of artificial intelligence in multi-objective optimization.

# RELATED WORKS

In this section, we provide an overview of the literature related to our work. We initially review prior studies that address the trade-off between accuracy and earliness in student dropout prediction (SDP). Concurrently, we examine reinforcement learning (RL) methods designed to balance these competing objectives.

## Methods for accuracy-earliness trade-off in SDP

There exists a very extensive literature on the topic of predicting student learning outcomes at the earliest with a reasonable level of accuracy in SDP task. The typical techniques found in the literature for addressing this problem could fall into two categories: post processing after prediction and processing in prediction.

The typical techniques found in the literature are post processing after prediction methods, which suggested training on complete time-series data and test on partial sequence data at different preset halting points. When the performance evaluation metric after prediction satisfies the preset stopping rules, it can serve as a basis for decision-making between earliness and classification accuracy. As an illustration, *Ben Soussia et al. (2022)* adopted the LSTM-Fix (*Ma, Sigal & Sclaroff, 2016*) approach, utilizing complete sequence data of student learning behaviors for classifier training. Subsequently, it calculates the harmonic mean (HM) (*Limbrunner, Vogel & Brown, 2000*) of predictive accuracy and earliness for partial sequence data at various preset halting points. Eventually, the HM value is compared to a preset threshold to signify an optimal early prediction time. *Dachraoui, Bondu & Cornuéjols (2015)* proposed a "NonMyopic" method to balance between early prediction and accuracy. It firstly clusters complete time series into prototypes. Then it calculates the mis-classification cost of partial sequence data in the testing phase based on their similarity to the prototypes. After that, the method recognizes the optimal warning time when the cost exceeds a preset threshold. While the post processing methods could make balancing decisions between the accuracy and earliness of prediction, they often carry a high computational burden (*Zhang et al., 2017*). Additionally, the reliance on fixed halting points and pre-defined thresholds lacks adaptability, failing to ensure the optimal trade-off across different learning behavior data (*Pachos et al., 2022*).

More sophisticated approaches have been developed by several authors using processing in prediction. These approaches pertain to real-time handling and balancing of conflicting objectives within the model's parameterized inference process. These existing methods can also be divided into two strategies. One involves explicitly introducing balancing constraints into the training algorithm to address the trade-off between conflicting objectives. For instance, *Deho et al. (2022)* analysis of student learning data over three years in an Australian university, incorporating a regularization term to balance model prediction fairness and efficacy. However, the selection and adjustment of regularization

parameters rely on characteristics of the specific problem, domain knowledge and data distribution, which is challenged to generalize across various scenarios. The other strategy utilizes MOO algorithms to mediate between competing objectives. Typically, *Jimenez et al. (2019)* proposed a temporal multi-objective optimization model to found the earliest horizon of a student's academic dropout. They incorporate an evolutionary algorithm to generate a Pareto front of non-dominated solutions for minimizing the waiting time while maximizing the predictive accuracy. An advantage of these approaches is that it leads to a solution set which will contain a policy for balancing between conflicting objectives. However, the non-convex nature of the solution set leads to a complex landscape of potential solutions, which makes it difficult to navigate the solution space and to effectively identify the global optimal solution (*Wang et al., 2023*).

### RL approaches for accuracy-earliness trade-off

Reinforcement learning algorithms excel in dynamic adaptation and real-time optimization. This makes them highly effective for complex decision-making scenarios and particularly adept at solving multi-objective optimization problems where conflicting goals are present (*Ge et al., 2022*; *Yang et al., 2023*; *Zhou, Du & Arai, 2023*). Hence, some studies have incorporated reinforcement learning into the decision-making process for early classification of time series (*Martinez et al., 2018*; *Martinez et al., 2020*; *Hartvigsen et al., 2019*).

The core idea of these methods is to train an early classifier using partial sequence data. To be specific, they treat partial sequences as environmental states, and employ a reinforcement learning agent to decide whether classifying incomplete sequences immediately or waiting to gather more information. By maximizing reward functions and updating strategies in real-time, these approaches allow models to search the solution space more efficiently and learn an decision strategy to find the optimal balance in dynamic environments.

Its current form can largely be attributed to the work of *Martinez et al. (2018)*, which developed an end-to-end early classifier called ECTS under the RL framework, aiming at finding a compromise between classification accuracy and earliness. They encoded the competing objectives into several reward functions. The objective of accuracy is shaped by a scalar reward. When the classifier correctly classifies the time series, the agent gets a positive reward. In contrast, an incorrect prediction will lead to a negative reward. The objective of earliness is encoded by a negative reward proportional to time, *i.e.,* the longer the wait, the greater the negative reward. The trade-off between the two competing objectives is carried on a user controlled hyper-parameter. Subsequently, they utilized a deep neural network as a function approximator of the Q-function, and adapted the Deep Q-Network (DQN) algorithm (*Mnih et al., 2015*) to learn the optimal policy by maximizing the cumulative discounted reward. However, the classification actions of ECTS occur less frequently than decisions to delay, leading to an uneven distribution of experience memory for the agent to learn from. Accordingly, *Martinez et al. (2020)* proposed a derivative work that addressed these issues through the use of a Double Deep Q-Network (DDQN) (*Van Hasselt, Guez & Silver, 2016*) paired with prioritized experience replay (PER) (*Schaul, Quan & Antonoglou, 2015*). In addition, they applied a monotonic non-decreasing function of time to shape

the negative rewards with each delayed decision. However, the trade-off user-defined hyper-parameters, imposing a significant burden on engineers to assign appropriate weights.

A new technique, similar in principle to *Martinez et al. (2018)* and *Martinez et al. (2020)* but using a different underlying technique was proposed by *Hartvigsen et al. (2019)*. They proposed a novel model called EARLIEST, which leverages a joint-optimization solution by combining the goals of accuracy and earliness into a unified objective function. On the one hand, to achieve the goal of accuracy, they manipulated a recurrent neural network (RNN)-based Discriminator network to capture complex temporal dependencies in time series, and adopted a cross entropy loss to minimize the Discriminator's errors. Meanwhile, they introduced a RL based stochastic Controller to learn a halting-policy for maximizing the performance of the Discriminator. They quantified the success of the Discriminator with a scalar reward and converted the maximization of the expected reward into the minimization of negative expectation. This transformation allows for unified model parameter optimization *via* gradient descent. On the other hand, the goal of earliness is determined by an additional loss term to encourage early halting, and the balance between contradictory goals depends on a hyper-parameter *lamda*.

Despite RL-based methods' ability to find compromised solutions between multiple objectives, they come with significant drawbacks. The primary concern is their reliance on scalarization to combine multiple objectives into one, often resulting in the loss of critical details about each objective's contributions and their interrelations (*Qin et al., 2021*). Additionally, the neglect of non-aligned optimal solutions limits the model's ability to adapt swiftly to changing preferences, leading to suboptimal decisions (*Basaklar et al., 2023*). Unlike previous RL methods, our approach addresses these issues by employing a vector reward mechanism that preserves the distinct contributions of each objective, enhancing the granularity of decision-making. Furthermore, the introduction of envelop Q-learning facilitates dynamic alignment with changing preferences, broadens the exploration of the solution space, and assists in identifying more accurate Pareto-optimal solutions.

# METHODOLOGY

## System model

Supposing we have a training dataset $\mathcal{D} = \left\{ \left( X^j, l^j \right) \right\}_{j=1..N}$ with $N$ pairs of complete temporal sequences $X$ and their associated label $l \in \mathcal{L}$, with $\mathcal{L}$ the set of labels. $X = (x_1, \ldots, x_T) \in \mathbb{R}^{T \times n}$ is the temporal sequence with maximal length $T \in \mathbb{N}^+$. At each time step $t \in [1, T]$, the measurement $x_t$ is a vector of $n \in \mathbb{N}^+$ features. We define a static classifier as a mathematical function $f_{\text{classif}}$ mapping from a temporal sequence $X$ to its label $l$ such that $f_{\text{classif}} : \{X\} \to \mathcal{L}$.

Given most of the relevant information comes from a small proportion of a time series, we assume that we could estimate the learning outcomes of students from partial sequences, and the class distribution of these partial sequences is independent and identically distributed (i.i.d.) with the complete sequences. The partial sequence is defined as $X_{:t} = (x_1, \ldots, x_t) \in \mathbb{R}^{t \times n}$ with $t \leq T$. Therefore the task of optimize the two competing scores of classification accuracy and earliness could be defined as a early classifier as

a mathematical function $f_{\text{early}} : \{X\} \rightarrow \mathcal{L} \times [1, T]$, which mapping a partial temporal sequence $X_{:t}$ to a label $l$ and predicting the optimal earliest time step $t^* \in [1, T]$ to perform classification:

$$t^* = \arg \max_{t \in [1, T]} Acc\left(f_{\text{early}}\left(X_{:t}\right), l\right) + Earliness(t).$$

Given the multi-objective nature of the problem, we are essentially dealing with a stochastic multi-objective optimization challenge. Our objective is to approximate the exact Pareto front (*Roijers et al., 2013*) by systematically searching for a set of policies, denoted as $\pi$, that collectively represent the optimal trade-offs between objectives.

## Preliminary
### *Markov decision process*

Optimizing the accuracy-earliness trade-off issue in student dropout prediction task fundamentally presents an multi-objective optimization challenge (*Jimenez et al., 2019*). Reinforcement learning, commonly employed in such contexts, tackles this by enabling an agent to learn through trial and error within an unknown environment (*Dulac-Arnold et al., 2020*). This learning process is typically modeled as a Markov decision process (MDP) (*Garcia & Rachelson, 2013*), which is defined by the tuple $\langle \mathcal{S}, \mathcal{A}, \mathcal{P}, r, \gamma \rangle$, where $\mathcal{S}, \mathcal{A}, \mathcal{P}$, $r$, and $\gamma$ represent state space, action space, transition distribution, reward and discount factor, respectively.

### *Scalar reward based RL*

In researches addressing multi-objective trade-off issues with reinforcement learning, scalarization is a prevalent technique where multiple objectives are combined into a single one through static weights (*Martinez et al., 2018*; *Martinez et al., 2020*; *Hartvigsen et al., 2019*). This process can be formalized as transforming multiple objectives into a single optimization goal through a scalarization function (*Zhang, Qi & Shi, 2023*). The transformation facilitates the application of standard reinforcement learning (RL) algorithms for finding a policy aimed at maximizing an agent's cumulative reward in a MDP. In this setting, the agent, defined by its policy $\pi$, chooses an action $a$ in each state $s$ as $a = \pi(s)$, with the environment responding by providing a reward $r = R(s, a)$ and transitioning to the next state $s' = \mathcal{P}(s, a)$. The The interaction sequence $\langle s, a, r, s' \rangle$ continues to unfold, with the goal of maximizing the cumulative reward, guiding the agent toward an optimal policy or a terminal state.

The agent's performance is quantified by the expected cumulative rewards, denoted by the action-value function, also known as the Q-function:

$$Q_{(s,a)} = \mathbb{E}_{\pi}\left[\sum_{k=0}^{\infty} \gamma^k r_{t+k} | s_t = s, a_t = a\right]. \tag{1}$$

The Q-function indicates, for a given policy $\pi$, if selecting an action $a$ in a particular state $s$ is likely to have good repercussions in the following steps by getting large rewards or not.

The optimization process for standard value-based MDPs incorporates the utilization of Bellman's optimality equation. This equation facilitates the decomposition of the action value into the immediate reward added to the discounted action value of the subsequent state.

$$Q^*(s,a) = r(s,a) + \gamma \mathbb{E}_{s' \sim \mathcal{P}(\cdot|s,a)} \max_{a' \in \mathcal{A}} Q^*(s',a'). \tag{2}$$

The loss function used for updating can be calculated by:

$$
\begin{aligned}
L(\Theta) \quad &= \left(Q^*(s,a,\Theta^-) - Q(s,a,\Theta)\right)^2 \\
&= \left(r + \gamma \operatorname*{argmax}_a Q(s',a,\Theta^-) - Q(s,a,\Theta)\right)^2.
\end{aligned}
\tag{3}
$$

## Problem reformulation

Diverging from the conventional RL methods that formulate early classification of time series within MDP, our approach innovatively reformulates the conflicting objectives within a Multi-Objective MDP (MOMDP) framework. This MOMDP is comprehensively characterized by the tuple $\langle \mathcal{S}, \mathcal{A}, \mathcal{P}, \mathbf{r}, \gamma, \Omega \rangle$. It encompasses a state space $\mathcal{S}$, an action space $\mathcal{A}$, and a transition distribution $\mathcal{P}(s_{t+1}|s_t, a_t)$. Significantly, the model incorporates a reward vector $\mathbf{r}(s_t, a_t)$ and a preference space $\Omega$, allowing for a nuanced and holistic representation of the complex interplay between various objectives in time series classification.

Each element of the tuple is defined as follows.

1. States space

   In the task of accuracy-earliness trade-off for SDP, the objective is to predict labels $l \in \mathcal{L}$ as early as possible, without access to complete sequence information. Consequently, at each timestep $t$, the state $s_t$ is represented by a set of partial time series variables $X_{:t}$, essentially a slice across all variables at timestep $t$. The partial sequence is a key aspect of the problem, reflecting the real-world challenge of making early predictions based on limited information.

2. Action space

   The action space is $\mathcal{A} = \mathcal{A}_c \cup a_d$. If the agent selects $a_d$, it signifies the choice of the 'WAIT' action, leading to the advancement of the system by one timestep. The action selection process then restarts with the new state, $X_{t+1} = s_{t+1}$. On the other hand, if $\mathcal{A}_c = \mathcal{L}$ is choose, the agent opts for 'HALT', concluding the processing of the current time series and triggering a classification label prediction. The timestep at which 'HALT' is selected, or when $t$ reaches $T$ (the preset maximum limit), is identified as the halting point $\tau$. The action space is represented as follows:
   $$\mathcal{A} = \begin{cases} a_d, & \text{WAIT} \\ \mathcal{A}_c, & \text{HALT} \end{cases} \tag{4}$$

3. Reward vector

   Unlike the previous scalarization methods that use a single scalarized reward (*Martinez et al., 2018*; *Martinez et al., 2020*; *Hartvigsen et al., 2019*), we introduce a vectorized

reward. The difference between them is illustrated as Fig. 2. The reward vector can be represented as $\mathbf{r}(s_t, a_t) = [r_1(s_t, a_t), r_2(s_t, a_t)]$, for retaining crucial information about each objective. The first element of the reward vector, $r_1(s_t, a_t)$, pertains to the accuracy of predicted labels. The second element in the reward vector, $r_2(s_t, a_t)$, is associated with the earliness of prediction. To be specific, $r_1$ and $r_2$ are given as:

$$\mathbf{r}(s_t, a_t) = \begin{cases} [r_1 = 0, r_2 = -\lambda t^p] & , \text{if } a_t = a_d \\ [r_1 = 1, r_2 = 1] & , \text{if } a_t = l \\ [r_1 = -1, r_2 = 1] & , \text{if } a_t \neq l \end{cases} \tag{5}$$

When the agent opts to wait (denoted by action $a_d$), it receives no reward for accuracy ($r_1 = 0$) but incurs an increasingly negative reward over time for delay ($r_2 = -\lambda t^p$), with parameters $\lambda$ and $p$ determining the intensity and rate of this penalty. In contrast, if the agent makes a correct prediction (action $l$), it is rewarded on both accuracy and timeliness ($r_1 = 1, r_2 = 1$). However, an incorrect prediction penalizes accuracy ($r_1 = -1$) while still rewarding timeliness ($r_2 = 1$). This reward structure incentivizes the agent to make timely and accurate decisions, addressing the trade-off between accuracy and earliness.

4. Preference space

   To manage the complexity of the reward vector, we introduce the concept of preference space $\Omega$, which is typically represented as a vector $\boldsymbol{\omega} = (\omega_0, \omega_1)^\top \in \Omega$. It could be regarded as a series of rays extending radially in the first orthant in Fig. 1, being used to weigh the relative importance of different objectives encapsulated in the reward vector.

## Proposed framework

Within the framework of MOMDP, our model aims to obtain a single policy that covers the entire preference space for multiple conflicting objectives in a SDP problem. This scenario presents two primary challenges. The first is to identify a policy that approximates the global optimum for each optimization as closely as possible. Subsequently, the goal is to establish an optimization framework that is sufficiently efficient to allow for the optimization of a densely populated set of weight vectors.

As an attempt at solving these challenges, we initially sample preference vectors uniformly at random in each episode. This approach aims to comprehensively search the entire space of multiple objectives in a single invocation of optimization. Alongside this, we have designed a specialized deep Q-network to map the combined states and preference vector inputs into a multi-dimensional action-value matrix. This design is key in effectively disentangling and representing the action-values for various objectives. Additionally, we introduce the envelop Q-leaning (*Yang, Sun & Narasimhan, 2019*) to modify the standard Q-learning to be multi-objective. This modification enhances search efficiency by allowing the neural network to share representations across all combinations of weight vectors. The specific details of this approach are outlined as follows.

### *Learning the global optimal strategy*

In our approach, we train a deep Q-network to learn the global optimal policy. Specifically, at each step of the time series, we observe an environment state $s_t$ derived from the partial time series up to time $t$. This state is then combined with a preference vector $\boldsymbol{\omega} = (\omega_0, \omega_1)^\top$,

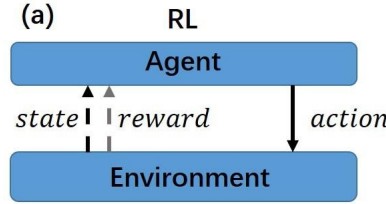

(a) RL ... Optimal action value function

$$Q^*(s,a) = max_\pi \boldsymbol{Q}(s,a)$$
$$\text{s.t. } \boldsymbol{Q}(s,a) = [Q_1(\text{s,a}), \cdots, Q_M(\text{s,a})]^T$$

(b) MORL ... Optimal vectorized action value function

$$Q^*(s,a) = max_\pi \boldsymbol{Q}^\pi(s,a)$$
$$\text{s.t. } \boldsymbol{Q}^\pi(s,a) = [\mathbf{Q}_1^\pi(\text{s,a}), \cdots, \mathbf{Q}_M^\pi(\text{s,a})]^T$$

**Figure 2 Comparison between RL and MORL settings.** (A) The traditional RL setting adopts a single scalar reward and the best action is determined by the maximum value among $M$ scalarized action-values, where $M$ represents the number of actions. (B) In the multi-objective setting, each objective corresponds to a reward signal, collectively forming a reward vector $\mathbf{r} = [r_1, r_2, \ldots, r_N]^T$, where $N$ is the number of objectives. Consequently, each element in the optimal vectorized action-value function becomes a vectorized Q-value, used to measure the value of actions across different objectives.

uniformly sampled from the preference space $\Omega : \sum_{i=0}^{L} \omega_i = 1$. The combination of $s_t$ and $\boldsymbol{\omega}$ creates a joint feature space, which serves as the input for our deep Q-network. The network is then trained on this composite input with the goal of discovering the global optimal policy. Conceptually, this training process can be visualized as exploring a series of rays extending radially in the first orthant of Fig. 1. The aim is to ascertain a Pareto set such that, corresponding to every valid $\omega_i$, there exists a point $d$ within the Pareto set for which the value of $\boldsymbol{\omega} \cdot \mathbf{F}(d)$ achieves maximization. In other words, for each scalarization direction indicated by a specific ray $\boldsymbol{\omega}$, Our target is to identify points within the objective space that are situated at the maximum distance from the origin along the direction of that particular ray.

Moving from the conceptual framework to the specific implementation, the architecture of our deep Q-network is detailed in Fig. 3. This architecture, comprising convolution layers, flatten layers, and multi-layer perceptron (MLP) layers. Its output, $\mathbf{Q}_\theta(s_t, \boldsymbol{\omega}, a_t)$, manifests as a probability matrix of dimensions $[1, M \times N]$, where M is the number of actions and N is the number of rewards. This output is then transformed into a matrix of dimensions $[M, N]$. Each row in this matrix corresponds to the action-value of each action, expressed as $\mathbf{Q_0}(s_t, \omega, a_t), \mathbf{Q_1}(s_t, \omega, a_t), \mathbf{Q_2}(s_t, \omega, a_t)$. The columns within this matrix represent the action-value for each objective, providing a detailed view of the potential outcomes for each possible action under different objectives.

Furthermore, to navigate the exploration-exploitation dilemma inherent in reinforcement learning, our global optimal strategy hinges on the $\epsilon$-greedy algorithm (*Sutton, 2020*) for allowing the agent to alternate between exploring new actions and exploiting known high-value actions. To be specific, at every timestep, the agent chooses a random action from the action space $\mathcal{A}$ with a probability $\epsilon$. Conversely, with probability $1 - \epsilon$, the agent opts for the action that maximizes the projected action-value

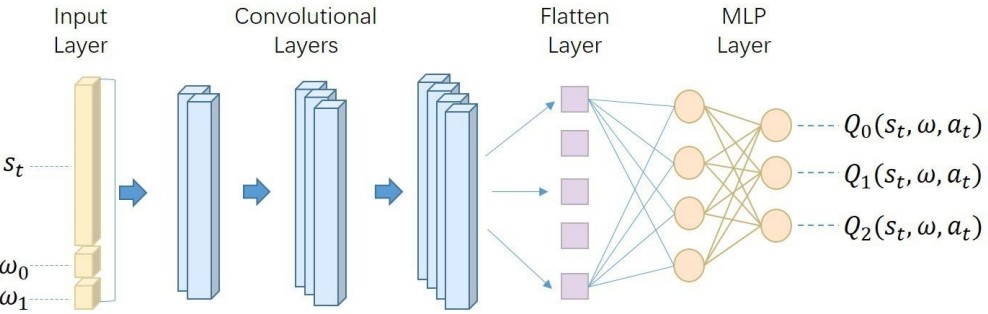

**Figure 3** **The network architecture of our deep Q-network.** It is meticulously designed to process inputs composed of the state representation $s_t$ and preference vector $\boldsymbol{\omega}$. This input is systematically mapped through a series of convolutional, flattening, and multi-layer perceptron (MLP) layers. The network's output is a set of action-values for each possible action, providing a comprehensive evaluation of each action's potential impact within the given state and preference context.

$\max_{a_t \in \mathcal{A}} \boldsymbol{\omega}^\top \mathbf{Q}(s_t, \boldsymbol{\omega}, a_t; \theta)$. This process involves:

$$a_t = \begin{cases} \text{random action in } \mathcal{A}, & \text{w.p. } \epsilon \\ \max\limits_{a_t \in \mathcal{A}} \boldsymbol{\omega}^\top \mathbf{Q}(s_t, \boldsymbol{\omega}, a_t; \theta), & \text{w.p. } 1 - \epsilon \end{cases} \tag{6}$$

where $\mathbf{Q}(s_t, \boldsymbol{\omega}, a_t; \theta) = (Q_0(s_t, \boldsymbol{\omega}, a_t; \theta), Q_1(s_t, \boldsymbol{\omega}, a_t; \theta), Q_2(s_t, \boldsymbol{\omega}, a_t; \theta))^\top$.

This approach not only ensures a robust balance between gaining new knowledge (exploration) and leveraging existing information (exploitation), but also aligns the agent's decisions with the multi-dimensional objectives in the SDP context, enhancing both the adaptability and effectiveness of the policy.

*Parameter training procedure*

Our training methodology for the deep Q-network is specifically tailored to the multi-objective challenges in student dropout prediction. Our approach significantly diverges from conventional reinforcement learning strategies which primarily rely on Q-learning algorithms. The Q-learning methods are underpinned by Bellman's optimality equation (as indicated in Eq. (2)) and focus on maximizing expected rewards through the iterative refinement of the agent's policy using temporal difference methods as Eq. (3). However, the standard Bellman's equation becomes inadequate in our multi-objective setting due to the introduction of preference vectors. To tackle this challenge, we adopt envelope Q-learning (*Yang, Sun & Narasimhan, 2019*), a refined version of traditional Q-learning specifically engineered for multi-objective optimization. This critical adaptation enables us to extend the DDQN into its multi-objective counterpart, MODDQN. The MODDQN framework, detailed in Algorithm 1 and visually represented in Fig. 4, integrating the principles of multi-objective optimization with the complexities inherent in SDP, ensuring that the policy not only aligns with various preference vectors but also adapts our concept of optimality based on vectorized rewards.

The training process begins with the initialization of the replay buffer $\mathcal{D}_\tau$, which is essential for storing transitions during the learning process. Concurrently, we initialize the

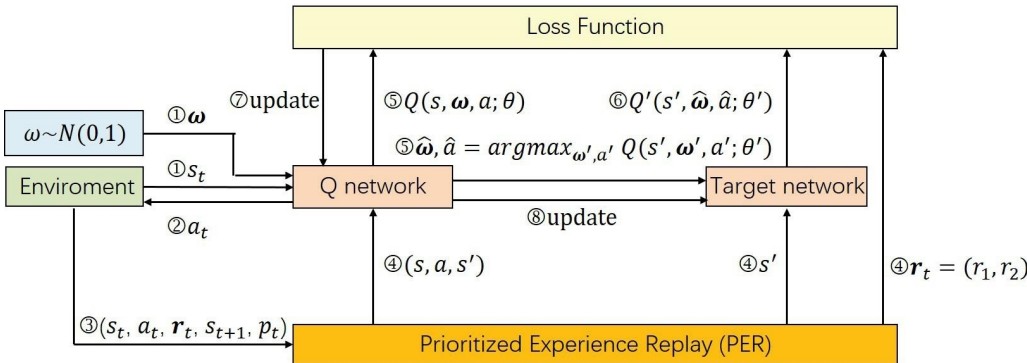

**Figure 4** **The parameter training procedure of our MODDQN algorithm.** The details are provided in Algorithm 1 as the form of pseudocode.

action-value function $\mathbf{Q}(s, \omega, a; \theta)$ and its counterpart, the target action-value function $\mathbf{Q}'(s, \omega, a; \theta')$, which is essential for calculating the temporal difference in reinforcement learning.

For every episode, and at each timestep within the episode's horizon, the model observes a state $s_t$, reflecting the current snapshot of a student's learning behavior sequence up to time $t$. It is concatenated with sampled preference vectors, and executing actions based on the $\epsilon$-greedy strategy as Eq. (6). The agent's action leads to an environmental response, providing the next state $s_{t+1}$, a multi-objective reward vector $\mathbf{r}_t$ and the terminal indicator. Each transition, encapsulating the state, action, reward vector, next state and terminal indicator is stored in the replay buffer. When a mini-batch of the stored transitions is accumulated, they are utilized in the envelop Q-learning through the experience replay mechanism.

The envelope Q-learning, pivotal in our MODDQN framework, revolutionizes traditional Q-learning for multi-objective optimization. By employing vectorized value functions and updating them *via* the convex envelope of the solution frontier, this method deftly navigates the intricate landscape of multi-objective optimization. This is in stark contrast to scalarized Q-learning, which tends to optimize policies in isolation for each preference. Instead, envelope Q-learning concurrently optimizes a spectrum of policies across various preferences. This alignment with our multi-dimensional concept of optimality in SDP is concisely captured in the evaluation operator equation:

$$Q^*(s, a, \omega) = \mathbf{r}(s, a) + \gamma \, \mathbb{E}_{s' \sim \mathcal{P}(\cdot | s, a)} \max_{a' \in \mathcal{A}} Q^*(s', a', \omega). \tag{7}$$

By adopting this approach, we ensure that our policy is not just responsive but intricately aligned with the diverse and often conflicting objectives inherent in the realm of Student Dropout Prediction.

Furthermore, due to the imbalance in the agent's memory—where classification actions are less frequent compared to delay actions—we adopt the Prioritized Experience Replay (PER) method. PER enhances learning efficiency from significant experiences and ensures more relevant transitions are frequently replayed. This method prioritizes transitions with

higher learning potential, leading to a more focused learning process. To be specific, the calculation of the priority in PER for the $l$th transition is executed by:

$$p_l = |\delta_l| + \chi \tag{8}$$

where $\delta_l$ denotes the temporal difference (TD) error of the $l$th transition, and $\chi > 0$.

After calculating the priority of the $l$th transition, the transition is stored in memory buffer $\mathcal{D}_\tau$ as $(s_t, a_t, \mathbf{r}_{t+1}, s_{t+1}, p_l)$. When the total number of combined transitions exceeds the capacity of $\mathcal{D}_\tau$, the training of the model can proceed through the sampling of a mini-batch, guided by the probability distribution outlined below:

$$P_l = \frac{p_l^\alpha}{\sum_m p_m^\alpha} \tag{9}$$

where $P_l$ is proportional to the priority $p_l$ of a transition, and $\alpha$ signifies how much prioritization being used.

Finally, the deep Q-network is updated using MODDQN with PER, and the loss function employed for updating the deep Q-network can be determined by:

$$L_1(\theta) = \mathbb{E}_{s,a,\boldsymbol{\omega}} \left[ \| \mathbf{y} - \mathbf{Q}(s, a, \boldsymbol{\omega}; \theta) \|_2^2 \right] \tag{10}$$

where $\mathbf{y}$ is given by:

$$\mathbf{y} = \mathbb{E}_{s'} \left[ \mathbf{r} + \gamma \mathbf{Q}' \left( s', \underset{\boldsymbol{\omega}', a'}{\arg\max} \, \omega^\top \mathbf{Q}\left(s', \boldsymbol{\omega}', a'; \theta\right); \theta' \right) \right]. \tag{11}$$

Given optimizing $L_1(\theta)$ directly poses practical difficulties as the optimal frontier encompasses a vast array of discrete solutions, rendering the loss function's landscape significantly non-smooth. To mitigate this, an auxiliary loss function $L_2(\theta)$ is employed:

$$L_2(\theta) = \mathbb{E}_{s,\boldsymbol{\omega},a} \left[ \| \boldsymbol{\omega}^\top \mathbf{y} - \boldsymbol{\omega}^\top \mathbf{Q}(s, \boldsymbol{\omega}, a; \theta) \|_2^2 \right]. \tag{12}$$

Ultimately, the target network $Q'$ undergoes updates at every $T_{update}$ steps. The final loss function is formulated as follows:

$$L(\theta) = L_1(\theta) + L_2(\theta). \tag{13}$$

Within the framework of Multi-Objective Reinforcement Learning algorithm, $L_1$ initially ensures that the Q-value is approximated to any real expected total reward, despite potential challenges in achieving optimality. Subsequently, $L_2$ exerts an auxiliary influence, nudging the current estimate towards a direction of enhanced utility. Following the learning phase, the agent is equipped to adapt to any given preference by merely inputting $\boldsymbol{\omega}$ into the network, and our model could straightforwardly select the action with the highest Q-value as determined by the policy $\Pi_{\mathcal{L}}(\boldsymbol{\omega})$.

## EXPERIMENT

### Dataset description

In our study, we leverage two benchmark datasets to evaluate the efficacy of our proposed model, both of which are obtained from XuetangX, the largest Massive Open Online Course

---

**Algorithm 1** Training Algorithm

---

1: Initialize replay buffer $\mathcal{D}_\tau$.
2: Initialize action-value function $\mathbf{Q}(s, \omega, a; \theta)$.
3: Initialize target action-value function $\mathbf{Q}'(s, \omega, a; \theta')$ by copying: $\theta' \leftarrow \theta$.
4: **for** episode $= 1, \ldots, M$ **do**
5:     Sample a training pair from the dataset $\{(X^i, y^i)\}_{i=1\ldots n}$
6:     **while** not *terminal* and $t \leq T$ **do**
7:         Obtain partially observable state $s_t = X^i_{:,t}$.
8:         Sample a preference $\omega \sim \mathcal{D}_\omega$ and concatenate it with state $s_t$.
9:         Agent receives the input $[s_t, \omega]$ and picks an action $a_t$ based on Eq (**??**).
10:         Environment steps forward according to $a_t$ and gets the multi-objective reward vector $\mathbf{r}_t$, the next state $s_{t+1}$, and the terminal state.
11:         Store transition $(s_t, a_t, \mathbf{r}_t, s_{t+1}, terminal)$ in $\mathcal{D}_\tau$.
12:         **if** *update* **then**
13:             Sample $N_\tau$ transitions $(s_j, a_j, \mathbf{r}_j, s_{j+1}) \sim \mathcal{D}_\tau$ according to Eq (**??**).
14:             Sample $N_\omega$ preferences $W = \{\omega_i \sim \mathcal{D}_\omega\}$.
15:             Compute $\mathbf{y}$ according to Eq (**??**).
16:             Compute the loss function based on Eq (**??**) and Eq (**??**).
17:             Update Q-network by minimizing the loss function according to Eq (**??**).
18:         **end if**
19:         **if** $a_t = $ WAIT **then**
20:             Increment time $t = t + 1$.
21:         **else**
22:             Predict and set *terminal* = *True*.
23:         **end if**
24:     **end while**
25: **end for**

---

(MOOC) platform in China, accessible *via* https://www.xuetangx.com/. The initial dataset, known as KDDCup 2015, is available at https://www.biendata.xyz/competition/kddcup2015/data/. This dataset is widely recognized and utilized in the realm of MOOC dropout prediction research, with citations in works such as *Feng, Tang & Liu (2019)* and *Pulikottil & Gupta (2020)*. It comprises data related to 39 courses and 72,395 students, covered over a 30-day observation window and includes seven unique types of student learning activities, which are utilized as analytical features. The KDDCup 2015 dataset serves as a benchmark to compare our method against pre-existing techniques. The second dataset, also named XuetangX and of a larger scale, is found at http://moocdata.cn/data/user-activity. Initially, this dataset contained information on 246 courses and 202,000 students, spanning 22 types of events. To facilitate manageable model training, we implemented a data processing technique from *Prenkaj, Velardi & Distante (2020)*, which involved filtering out courses with less than 350 student trajectories. This process resulted in a dataset featuring 19 courses and 23,839 students, with the observation period extended to 35 days for the XuetangX

dataset. This refined dataset is used to test the robustness and generalization capability of our proposed model. Both the datasets have been anonymized, with UserIDs replacing usernames. This method of processing is common in student dropout prediction (*Prenkaj et al., 2020*; *Pan et al., 2022*), which can effectively address the concern on data privacy and ethical issues.

Guided by the procedures outlined in *Prenkaj, Velardi & Distante (2020)*, our analysis of each dataset involved the aggregation of student events on a daily basis to form temporal sequences. Through this approach, we were able to depict the activity of each student as a matrix, represented by $\mathcal{T}_u \in \mathbb{R}^{T,n}$. In this representation, $T$ denotes the duration of the selected observational window in days, while $n$ indicates the number of distinct event types present within the dataset. As a result, the final form of the input matrix for our analytical model is structured as a tensor with the dimensions $(N, T, n)$, where $N$ stands for the total student count.

Both datasets are partitioned following a distribution of 70% for training, 10% for validation, and 20% for testing. Given the presence of class imbalance within both datasets, we engage in downsampling of the training and validation sets to establish balanced subsets. Furthermore, we utilize the MinMaxScaler technique for normalization purposes. The test set is kept in its original form without any processing to maintain its authenticity and ensure the generalizability of our prediction results.

## Implementation details

In our MORL setting, the agent's action space includes three distinct options: wait, predict correctly, and predict incorrectly, therefore the dimension of action is 3. The reward is vectorized into two components, indicating prediction accuracy and earliness, making its dimension 2. The environment states are partial sequences extracted from the original time series up to a specified time step $t$. They are then extended with zero-padding to match the original sequence length, ensuring uniform dimensions. Consequently, the state size for the KDDCup2015 dataset is $1 \times 30 \times 7$, while for XuetangX, it measures $1 \times 35 \times 22$. Given the input of the Deep-Q Network is the combination of states and the two-dimensional preference vector $\omega$, we adjust $\omega$'s dimensions to facilitate the concatenation: for KDDCup2015, $\omega$ is expanded to $1 \times 30 \times 2$, and for XuetangX, to $1 \times 35 \times 2$. Post-concatenation, the Deep-Q Network's input is $1 \times 30 \times 9$ for KDDCup2015 and $1 \times 35 \times 24$ for XuetangX. We extract the joint feature *via* Conv1d, with three convolution layers and 128/256/128 kernels respectively. We set the kernel size to 3, stride to 1, and padding to 1. The convolution layers' output passes through a pooling layer and a fully connected layer to yield a Q-value with dimension of $1 \times 6$. Further, we reformat the Q-value to $3 \times 2$ to represent vectorized Q-values corresponding to the three actions. The learning process is optimized using the Adam optimizer with a learning rate set to $1e-3$. The hyperparameters $\lambda$ and $p$ that used in the reward vector for delay penalty are set to $1e-3$ and $1/3$, respectively. The model's exploration rate is initially set to 0.5, which is decayed to 0.05 after 500 iterations of learning. Other hyperparameters in our model include a memory size of 500, a batch size of 32 and $\gamma$ of 0.99. Our experimental code is

made publicly available, facilitating further exploration and replication of our study. The code can be accessed at https://github.com/leondepf/MORL-MOOC/tree/master.

## Baselines

In our study, we compared the MORL with models that are also designed to ensure prediction accuracy while achieving predictive earliness. The comparison approaches could be distributed into two types: non-reinforcement learning (Non-RL) methods and reinforcement learning (RL) methods. Non-RL Methods represent conventional techniques used in SDP for balancing between accuracy and earliness, which rely on fixed halting points and predefined threshold. In contrast, RL Methods dipict the trade-off approaches in the early classification of time series, which offer dynamic and adaptive strategies for early decision-making. This dichotomy allows us to evaluate the effectiveness of traditional methods against the more flexible, potentially more powerful reinforcement learning approaches.

1.  Non-RL methods

    - LSTM-Fix (*Ma, Sigal & Sclaroff, 2016*): LSTM-Fix involves training a classifier using the entire time series, but using only the initial part of the time series data available up to the preset halting point to make prediction. Its characteristic of relying on a predefined halting point for classification provides a contrast to MORL's dynamic early prediction capability.

    - NonMyopic (*Dachraoui, Bondu & Cornuéjols, 2015*): NonMyopic is chosen to showcase an approach that calculates optimal prediction timing of early warning for SDP, contrasting with MORL's method of learning from vectorized rewards to dynamically balance accuracy and earliness, highlighting the rapidity of MORL's prediction timing.

2.  RL methods

    - ECTS (*Martinez et al., 2018*): ECTS leverages a reinforcement learning framework to facilitate early classification of time series. It is conceptualized as a MDP, characterized by a scalar reward function and optimized by a DQN. This approach utilizes a user-preset parameter $\lambda$ to balance timely and accurate classification. By adjusting $\lambda$, users can finely tune the model to trade-off the dual objectives. Its comparison with MORL underlines MORL's superior handling of multi-objective optimization through MOMDP, vector reward function and MODDQN.

    - DDQN + PER (*Martinez et al., 2020*): It implements a DDQN with PER for demonstrating the effectiveness of using advanced reinforcement learning techniques to address the unbalanced memory issue. These advanced techniques are also adopted in our MORL model. The main difference between DDQN+PER and MORL lies in their reward design and optimization mechanisms, where DDQN+PER is based on scalar reward design and depends on preset hyper-parameter to combine different objectives, while MORL evolves vector reward and optimizes a single policy network across a spectrum of preferences without pre-setting them.

- EARLIEST (*Hartvigsen et al., 2019*): EARLIEST is a deep learning based method, which is composed of a RNN-based Discriminator with a RL-based Controller. A novel aspect of EARLIEST is the integration of minimizing Discriminator errors and maximizing Controller rewards into a unified loss function. Moreover, the model incorporates an additional loss term, regulated by the hyper-parameter $\lambda$, specifically designed to promote early halting. This innovative approach deftly merges the strategic decision-making prowess of reinforcement learning with the deep learning's predictive strengths. Its comparison to MORL highlights the latter's efficiency in navigating through multi-objective dilemmas using a single policy network, showcasing EARLIEST's complexity in managing similar tasks.

## Evaluation metrics

To comprehensively evaluate our model's performance, it is essential to consider a blend of metrics that collectively assess the trade-off between prediction accuracy and earliness, a core objective of the study. In this regard, we employ three key metrics: average accuracy (Avg. Acc), average proportion used (APU) and average harmonic mean (Avg. HM). Our selection for evaluation metrics is intentional and guided by the specific challenges of accuracy-earliness trade-off in SDP task. Avg. Acc is a standard metric, gauging the overall correctness of the model's predictions over testing data. However, in the context of accuracy-earliness trade-off in SDP, simply maximizing accuracy might encourage models to delay predictions until more data is available. Therefore, we introduce APU to specifically assess how early our model makes predictions, emphasizing the importance of timely interventions in educational settings. The Avg. HM combines both aspects, offering a single metric that encapsulates the trade-off between accuracy and earliness, which is the primary goal of our study. Together, these metrics provide a holistic evaluation framework, highlighting the suitability for the study's goals, a critical aspect not adequately captured by other common metrics in SDP.

1. Average accuracy

   Acc defines the model's average prediction accuracy on a testing set $\mathcal{D} = \left\{ \left( X^j, l^j \right) \right\}_{j=1..n}$ as:

   $$\text{Avg. Acc} = \sum_{j=1}^{n} \ell \left( f_{\text{classifier}} \left( X^j \right) = l^j \right) / n$$

2. Average proportion used

   A halting point $t_{\text{pred}}$ represents the earliest time step at which the agent decides to halt and predict a class label:

   $$t^j_{\text{pred}} = \min_{t \in [1,T]} \left\{ \underset{a \in \mathcal{A}}{\text{argmax}} Q \left( X^j_{:t}, a \right) \in \mathcal{A}_c \right\}$$

   Accordingly, the Average Proportion Used is computed as the mean of halting points on all sequences from the testing set, such that:

   $$\text{APU} = \sum_{j=1}^{n} t^j_{\text{pred}} / n$$

3. Average harmonic mean

   Avg. HM expresses the ability of our model to provide accurate predictions at the

earliest. The calculation of the Avg. HM is as follows:

$$\text{Avg. HM} = \frac{2*(1-\text{APU})*(\text{Avg. ACC})}{(1-\text{APU})+(\text{Avg. Acc})}.$$

## Results

### *Experimental comparison between RL methods*

In our comparative analysis, we first benchmarked our model against various reinforcement learning models, including EARLIEST, ECTS, and DDQN+PER. It is important to note the difference in training metrics: EARLIEST, being based on a deep learning approach, measures performance in epochs, indicating a complete cycle through the training dataset. In contrast, our model MORL, along with ECTS and DDQN+PER, are reinforcement learning models trained on episodes, each representing a full sequence of interactions in the environment.

To facilitate a consistent comparison across different training paradigms, we standardized our evaluation approach by focusing on iterations. Specifically, we assessed the models' performance during each parameter update in the model training's loss function process. This assessment was conducted on the testing set of both the KDD2015 and XuetangX datasets. The key metrics for evaluating model performance are Avg. Acc, APU and Avg. HM. We structured our assessment to provide outputs every 100 iterations, culminating in a total depth of 5,000 iterations. Furthermore, to ensure consistency in each code execution, we set a fixed random seed, stabilizing the initial random weights and data sequence, thereby maintaining uniformity in prediction results and loss. This method allowed for a detailed and consistent evaluation of models' performance across different datasets, ensuring comparability and robustness in our findings.

The experimental outcomes are illustrated in Figs. 5 and 6. Our model exhibits robust and stable performance on both KDDCUP2015 and XuetangX datasets, showing an increasing predictive accuracy with fewer time series data over iterations, especially in maintaining excellent harmonic scores. This trend signifies a consistent and effective balance between prediction accuracy and earliness, highlighting the model's proficiency in navigating the inherent trade-offs within these datasets. These findings highlight that our MORL model's strategy, which focuses on optimizing the convex envelope of multi-objective Q-values, ensures an efficient alignment between preferences and the corresponding optimal policies. This approach effectively tackles the challenge of optimizing multiple objectives simultaneously.

While the ECTS model occasionally surpasses our MORL model in predictive accuracy on the KDD15 dataset but generally underperforms on the XuetangX dataset. However, its performance shows significant fluctuations, likely due to the imbalanced agent's memory and the infrequency of classification actions, leading to uneven learning experiences.

Compared to ECTS, the DDQN+PER model demonstrates its stability, likely due to its implementation of prioritized sampling and prioritized storing. However, it still exhibits step-like fluctuations in predictive accuracy on both datasets and scores lower in harmonic metrics than our model. It may be attributed to the transformation of the multi-objective problem into a single-objective one, which cannot be tailored to optimize for certain preferences.

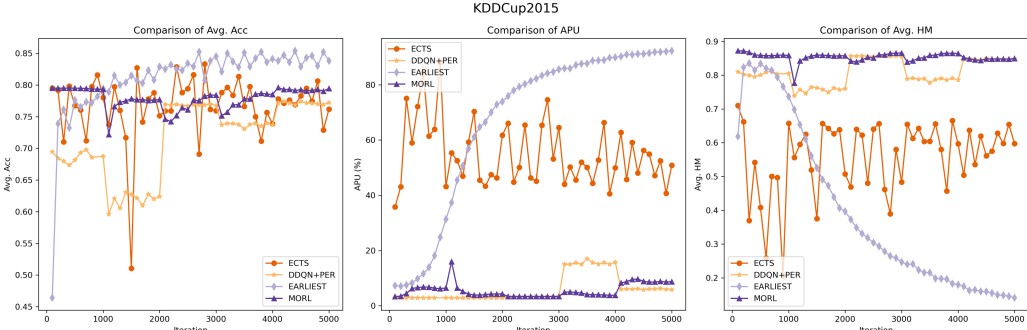

**Figure 5** **Performance comparison of the MORL algorithm against other RL based models on the KD-DCUP2015 dataset.** The purple line represents the MORL algorithm's performance across different iterations. As reflected by the comparison of *harmonic_mean_val*, demonstrating MORL's ability to achieve an optimal trade-off between prediction accuracy and earliness.

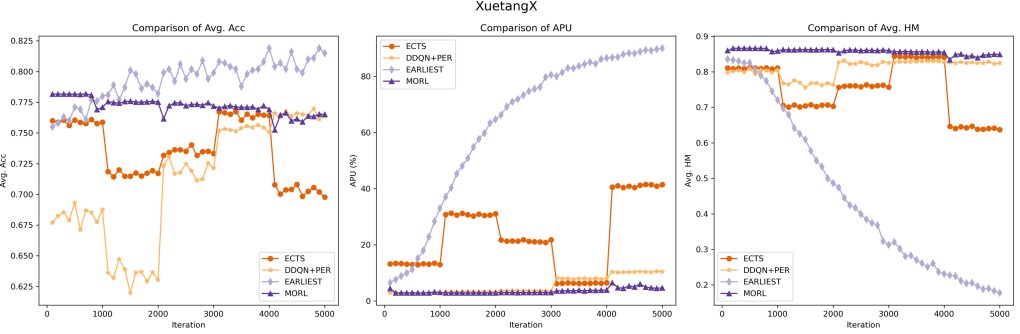

**Figure 6** **Performance comparison of the MORL algorithm against other RL based models on the Xue-tangX dataset.** Despite our MORL model indicated by the purple line exhibits a slightly lower *acc_val* compared to few baselines, it achieves the highest Avg. HM score across various iterations, underscoring the MORL algorithm's consistent and superior performance in balancing accuracy and earliness.

The experimental results for the EARLIEST model were somewhat unexpected. Although it demonstrates stability and gradual improvement in predictive accuracy across iterations, this approach requires more data in the later stage of the time series to maintain the prediction accuracy. This situation could result in compromises to prediction accuracy, leading to a decrease in its harmonic score. The experimental results suggest that, despite utilizing reinforcement learning for early stopping decisions, the EARLIEST model's approach of using an additional loss term and preset hyper-parameters to balance prediction accuracy and earliness fails to achieve a satisfactory multi-objective balance in certain specific datasets.

### Experimental comparison between Non-RL methods and RL methods

We conducted a comprehensive comparison of our MORL model against various Non-RL and RL methods. Our goal was to identify which method achieves the optimal trade-off between prediction accuracy and earliness on the KDD2015 and XuetangX testing sets.

**Table 1  Comparison results on KDDCup2015 and XuetangX datasets.**

| Dataset | Category | Models | Avg. Acc ↑ | APU ↓ | Avg. HM ↑ |
|---|---|---|---|---|---|
| KDDCup2015 | Non-RL methods | LSTM-Fix | **0.8393** | 100% | 0.0641 |
| | | NonMyopic | 0.7753 | 18.308% | 0.7956 |
| | RL methods | ECTS | 0.7590 | 44.734% | 0.6396 |
| | | DDQN+PER | 0.7369 | 15.03% | 0.7893 |
| | | EARLIEST | 0.8380 | 92.322% | 0.1407 |
| | | MORL (ours) | 0.7945 | **8.717%** | **0.8496** |
| XuetangX | Non-RL methods | LSTM-Fix | 0.8127 | 100% | 0.0552 |
| | | NonMyopic | 0.7670 | 8.741% | 0.8335 |
| | RL methods | ECTS | 0.7451 | 43.162% | 0.6449 |
| | | DDQN+PER | 0.7611 | 10.526% | 0.8225 |
| | | EARLIEST | **0.8470** | 95.352% | 0.0881 |
| | | MORL (ours) | 0.7770 | **4.283%** | **0.8577** |

**Notes.**

↑ Indicates that the higher the value, the better the performance, while ↓ represents the lower the better. Of all the results, the highest are shown in bold. The second highest results are shown with underlines.

For Non-RL methods, we began by evaluating the LSTM-Fix model, trained using the complete sequence of the training set. This model's predictive accuracy was assessed at predefined static halting points. We noted the highest predictive accuracy and the corresponding proportion of the sequence utilized. The NonMyopic model, claiming to predict the optimal time of early warning for student dropout, was trained using the full sequence. Its predictive accuracy was then evaluated at the most appropriate early classification time on the testing set.

In the RL category, we selected optimal policies that performed best during training, *i.e.,* among the most accurate policies, we selected the quickest one. We then evaluated this selected model on the testing set across five test trials. We reported the Avg. Acc and the APU on both MOOC datasets. These evaluation metrics were subsequently used to compute the Avg. HM, offering a balanced measure of accuracy and earliness. The results of these comparisons are summarized in Table 1.

The experimental results indicate that our model can achieve comparable prediction accuracy with the least amount of data on two datasets when compared to other models. This enables our model to obtain the highest harmonic mean, affirming its superior performance in achieving a favorable trade-off between prediction accuracy and earliness. It may attributed to the ability of MORL's optimization over the space of all possible preferences could quickly align one preference with optimal rewards and produce the optimal policy for any user-specified preference.

LSTM-Fix and EARLIEST alternately achieved the first and second highest prediction accuracy on both datasets. However, their high accuracy was attained nearly using the whole sequence. This reliance on sacrificing earliness to gain prediction accuracy does not effectively balance the conflicting accuracy-earliness objectives in the SDP task. This demonstrates that both of these methods could only care about one objective and face challenges in balancing multiple conflicting objectives.

The NonMyopic model, which claimed could calculate optimal prediction timing for early classification, is on par with the predictive accuracy with MORL, but utilized significantly more data to reach the same performance. This may account for the static method can also achieve predictions as early as possible while ensuring prediction accuracy to a certain extent, but it may fail to dynamic adaptation to related tasks with different preferences.

The ECTS and DDQN+PER models underperform the MORL model across three evaluation metrics, indicating that the dependence on scalar reward to combine different objectives could only learn an sub-optimal policy over the space of preferences but cannot be tailored to be optimal for specific preferences. By comparison, the MORL model can effectively use the information of $\max_a Q(s, a, \omega')$ to update the optimal solution aligned with a different preference $\omega$ in the multi-objective space. It is also noted that the ECTS model requires 30% more time series data than the DDQN+PER model to achieve comparable prediction accuracy, further confirming that the DDQN+PER model addresses the issue of memory imbalance in the ECTS model through the DDQN and PER methods. The techniques were also adopted in our MORL model.

### Policy adaptation

During the adaptation phase, we assessed how our MORL agent responded to user preferences on the XuetangX dataset. The experimental setup included a dynamically adapting MORL agent and three control groups with fixed preference settings, where the weights of accuracy (Acc) and earliness were set at ratios of 0.3:0.7, 0.5:0.5, and 0.7:0.3, respectively. All models underwent training across 10,000 episodes, with their parameters saved for subsequent analysis.

During the testing phase, we varied the weight of the preference on prediction accuracy from 0.1 to 0.9. We assessed the model's predictive Avg. Acc, APU and Avg. HM across different Accuracy ratios. Each data point was computed using three tests. We then plotted comparative curves of these three metrics (refer to Fig. 7). In these plots, the solid lines represent the mean of three independent runs under each configuration, whereas the light shadow denotes the standard deviations..

The results clearly demonstrate that our MORL agent consistently achieved robust predictive outcomes across a variety of preference ratios. This highlights the agent's capability to adeptly adjust to the user's preference. Furthermore, our deep MORL algorithm consistently outperformed other algorithms, particularly in scenarios where success was deemed more critical to the user.

In contrast, models trained with fixed preferences, although capable of achieving high predictive accuracy at higher Acc ratios, often failed to maintain this performance at lower Acc ratios. This limitation may stem from the fact that fixed preferences only find optimal solutions along predefined scalarized directions. When tested with varying preference ratios, the choice of weights might bias the model towards certain regions of the objective space. Consequently, this approach might overlook other potential optimal solutions not aligned with these predefined directions, leading to local, rather than global, Pareto optimality.

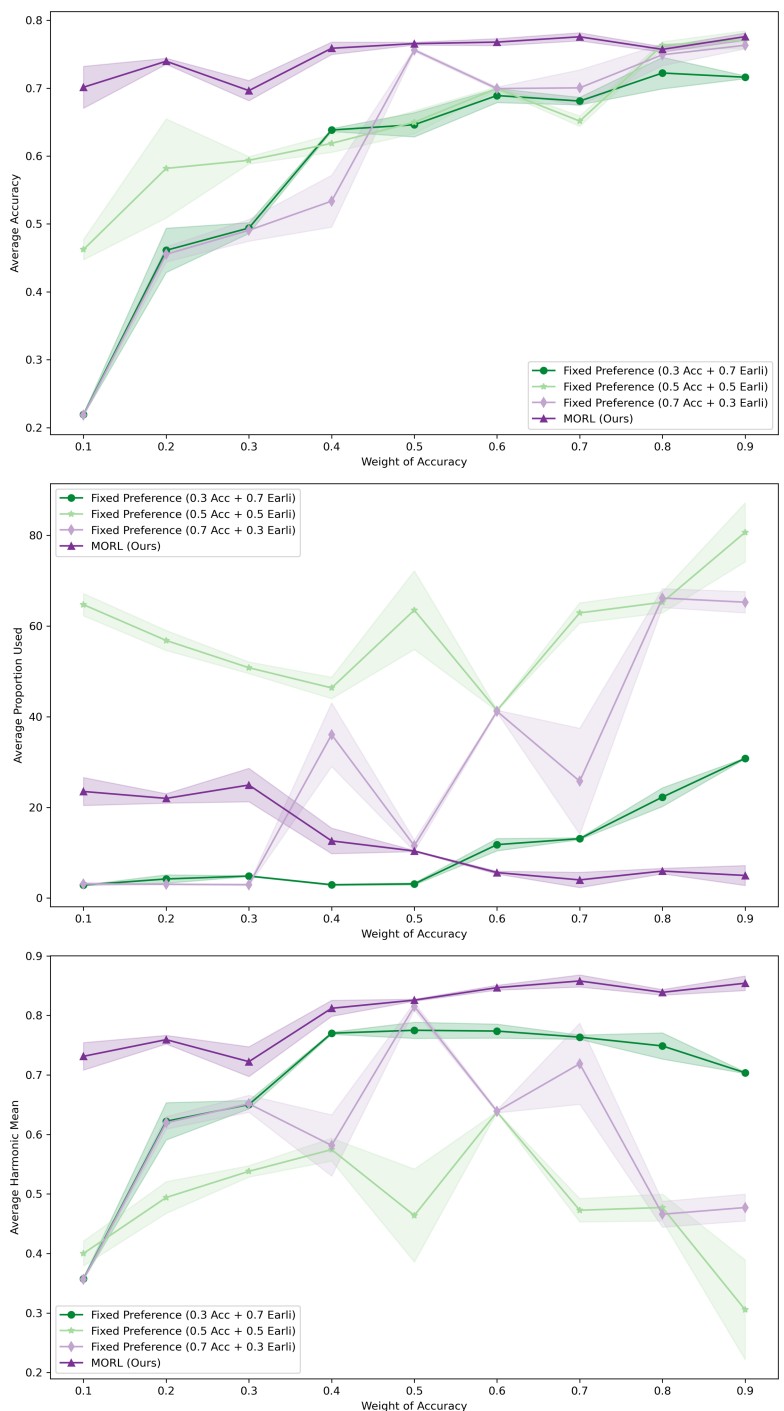

**Figure 7 Comparative performance of MORL agent across varying user preferences in XuetangX dataset.** It exhibits robust predictive outcomes across a variety of preference ratios and can be adaptive to different user's preference.

## CONCLUSIONS

In this study, we have broken new ground with a MORL methodology tailored for navigating the accuracy-earliness trade-off in SDP, addressing the complexities of optimizing multiple objectives simultaneously. Our approach ingeniously incorporates multiple reward perspectives and vectorized value functions, preserving the distinct contributions of each objective. This strategy effectively circumvents the information loss typically associated with scalarization methods. Through the innovative use of envelope value updates, our model is designed to flexibly adjust to user-defined preferences, ensuring the delivery of an optimal policy that skillfully manages the balance between accuracy and earliness, where it demonstrates marked superiority over contemporary state-of-the-art models, especially in metrics such as average proportion used and average harmonic mean. This research not only paves the way for more informed and effective decision-making in student interventions but also sets a precedent for the application of MORL in complex decision-making scenarios.

However, a primary limitation of our approach is its reliance on static weights for each objective throughout the training phase. Future research should explore the adoption of dynamic weights that evolve over time, allowing for a more flexible adaptation to changing objectives. Additionally, our current model operates with a single agent, reflecting a single-policy, preference-based framework. Future endeavors should extend to a multi-objective, multi-agent reinforcement learning (MOMARL) paradigm, enabling the generation of multiple solutions catering to various objective preferences through cooperative decision-making among agents. Moreover, the exploration of hyperparameter optimization within a hierarchical reinforcement learning context, where an auxiliary agent could sequentially select hyperparameters, represents a promising avenue for further research. These advancements are crucial for deepening the understanding and applicability of MORL in progressively more complex and dynamic scenarios.

## ACKNOWLEDGEMENTS

The authors are grateful to Chunhong Zhang and Benhui Zhuang for providing insightful feedback on earlier drafts of this article. We also want to thank anonymous reviewers and editors for providing helpful comments. Additionally, we would like to gratefully acknowledge the organizers of KDD Cup 2015 and XuetangX for making the datasets available.

### Funding

The authors received no funding for this work.

### Competing Interests

The authors declare there are no competing interests.

## Author Contributions

- Feng Pan conceived and designed the experiments, performed the experiments, analyzed the data, performed the computation work, prepared figures and/or tables, authored or reviewed drafts of the article, and approved the final draft.
- Hanfei Zhang conceived and designed the experiments, performed the experiments, performed the computation work, prepared figures and/or tables, authored or reviewed drafts of the article, and approved the final draft.
- Xuebao Li analyzed the data, authored or reviewed drafts of the article, and approved the final draft.
- Moyu Zhang analyzed the data, authored or reviewed drafts of the article, and approved the final draft.
- Yang Ji conceived and designed the experiments, authored or reviewed drafts of the article, and approved the final draft.

## Data Availability

The MORL-MOOC dataset is available at GitHub: https://github.com/leondepf/MORL-MOOC/tree/master.

## Supplemental Information

Supplemental information for this article can be found online at http://dx.doi.org/10.7717/peerj-cs.2034#supplemental-information.

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
