# Peer review of "Achieving optimal trade-off for student dropout prediction with multi-objective reinforcement learning"

_PeerJ Computer Science, doi:10.7717/peerj-cs.2034_

## Round 0.1 · original submission · Major Revisions

· Academic Editor

Major Revisions

The reviewers have given detailed comments regarding this paper, which helps improve the overall quality. In summary, they think that the authors should have a clear articulation of experimental assumptions and limitations, along with improved comparative analysis with existing methods. The methodology requires reorganization for clarity and coherence, and explanations regarding the choice of metrics and hyperparameters need enhancement. Additionally, clarity is required on the post-training decision-making process and the relationship between λ parameters in equations. While the findings are valuable, certain discrepancies in MORL's performance need addressing, particularly regarding its impact on Avg. ACC. The conclusion should explicitly address the potential limitations of MORL and outline avenues for future research. Further exploration of ethical considerations and biases within the model would strengthen the paper's validity. Also, there is a reviewer who thinks that the current title should be revised. The author should provide detailed responses to their comments and prepare a major revision of this paper. The quality should be further improved.

**Language Note:** The review process has identified that the English language must be improved. PeerJ can provide language editing services - please contact us at [email protected] for pricing (be sure to provide your manuscript number and title). Alternatively, you should make your own arrangements to improve the language quality and provide details in your response letter. – PeerJ Staff

Reviewer 1 ·

Basic reporting

The manuscript introduces a Multi-Objective Reinforcement Learning (MORL) algorithm to tackle the inherent challenge of balancing conflicting optimization requirements for Accuracy and Earliness in predicting student dropout on online education platforms. Experimental results demonstrate the superiority of the proposed MORL over baseline methods. The MORL approach excels by modeling the Student Dropout Prediction (SDP) problem as a novel multi-objective Markov Decision Process and leveraging the Double Deep Q-Network (DDQN) for learning the optimal policy. The manuscript effectively motivates and contributes to addressing this issue. However, certain improvements are needed, such as enhancing the clarity of the Introduction section to vividly illustrate the challenge of balancing prediction accuracy and earliness in the education domain. Additionally, attention to writing and grammar is required, with specific mention of improving the presentation of Figure 2's caption. The manuscript would benefit from the inclusion of more closely related references.

Experimental design

Clearly articulating the assumptions made during the experiments and addressing the limitations of the proposed methods is crucial for transparency and a comprehensive understanding of the study.

If the experiments do not explicitly showcase the Pareto Frontier, it may be advisable to exclude its detailed definition from the methods section and instead reference it through citations, streamlining the content.

Emphasizing the distinctive features of each baseline in contrast to the MORL algorithm, both in terms of methodology and expected outcomes, would enhance the clarity and comparative analysis.

Post-training, it is unconventional to utilize a greedy algorithm for determining actions. Typically, decisions involve directly selecting the action with the highest value output by the policy. This nuance should be explicitly stated to accurately represent the decision-making process employed in practice.

The Proposed Framework section requires reorganization to enhance clarity and coherence in presenting the methodology.

Validity of the findings

The findings are generally valuable and persuasive, contributing to the overall strength of the study.

In the Experiments section, Figures 5 and 6 illustrate the performance of different algorithms, where the purple line signifies the MORL algorithm. Notably, there seems to be a potential discrepancy in the MORL algorithm's performance in acc_val compared to expectations.

The conclusion section should explicitly address the potential limitations of the proposed MORL algorithm and outline avenues for future research, providing a comprehensive perspective on the study's implications and potential areas for improvement.

Further connections could be established between the algorithms and application scenarios.

Additional comments

1) The authors might want to polish the paper and make the English presentation more concise and accurate.

2) The manuscript exhibits several instances of incorrect word / definition usage. A comprehensive review and revision of the text are recommended to rectify these errors. Specifically, on line 391, the symbol "χ ¿0 " appears to be inaccurately presented. Additionally, on line 402, the term "hyper-parametric" may be incorrect.

3) The authors might want to double check the format and accuracy of the references.

Reviewer 2 ·

Basic reporting

An interesting study. The paper is well-structured, with a clear introduction to the problem, an overview of related work, detailed methodology, comprehensive experimentation, and conclusive findings. It articulates the significance of balancing accuracy and earliness in dropout predictions and introduces an innovative MORL strategy to address this issue. However, it does not fully meet PeerJ’s publication criteria as it currently stands. Therefore, the manuscript should address the following points before being accepted for publication.
1 - The title seems too long. Consider shortening it for clarity and impact.
2 - The literature review is thorough but could be expanded to include a broader range of studies on predictive modeling in education, providing a more extensive background against which to situate the current research.
3 - Figure 4/5/6 should benefit from further clarification or more detailed captions to enhance their interpretability for readers.
4 - The connection between results and hypotheses may not be explicitly clear in all sections.
5 - In Lines 259 – 260, the symbols of transition distribution and reward vector are inconsistent.

Experimental design

1 - The paper should clearly define its research question in the introduction.
2 - It's crucial to explain how each baseline contributes to a comprehensive evaluation framework, showcasing the proposed method's advantages, limitations, and innovative aspects in relation to existing approaches.
3 - The statement in the Preliminary about NP-hard may lead to confusion without further explanation. It is important to clarify why this problem is categorized as NP-hard.
4 - In Equation 5, what is the design principle behind the accuracy reward and earliness reward?
5 - Methods are described, but some sections may lack the detail necessary for replication.

Validity of the findings

The proposed method does not significantly outperform the baselines in Table 1, especially in the metric Avg. ACC. If the performance on the Avg.HM are believed to be benefited by MORL, then, why the Avg. ACC might be harmed by MORL, which make the overall performance sub-optimal?

Additional comments

Considering the innovative approach and the significant advancements demonstrated through comprehensive evaluations, I recommend the paper for minor revisions.

Reviewer 3 ·

Basic reporting

The paper presents a novel Multi-Objective Reinforcement Learning approach for Student Dropout Prediction that aims to balance accuracy and earliness without preset hyperparameters. It introduces a vector reward mechanism within a Multi-Objective Markov Decision Process (MOMDP) framework, leveraging Double Deep Q-Network (DDQN) and envelope Q-function updates. The article is a valuable and original scientific contribution, so I must congratulate the authors for their great work. However, it would be advisable to resolve some shortcomings before being published:
1. I suggest that the authors further sort out the logic of expression.
2. Your methodology would benefit from a more explicit comparative analysis with existing methods. This includes discussing why MORL was chosen over other potential approaches and its specific benefits in the context of student dropout prediction.
3. Research questions, that drive the paper, should be built in the introduction from an ongoing and pertinent bibliography (up to 2023). These should be of global interest and not focused to a particular local problem. Identifying a research gap is not enough; key is showing its significance to the field.
4. In the comparison of methods within the Related Works section, it would be beneficial to emphasize their application in the education domain. This focus can help contextualize the significance of your work and its contributions to this specific area.
5. The explanation in Figure 2 appears to be limited, potentially leaving readers without a solid grasp of the significance and implications of this transformation.

Experimental design

1. Discuss any assumptions made in the introduction and their potential impact on the generalizability of the results.
2. In Eq(13), the authors have introduce a hyper-parameter to optimize the loss fuction L(θ). But the paper lacks a detailed discussion on the method used for tuning λ. The authors should introduce how sensitive is the model to the choice of hyperparameters, and what strategies are recommended for their selection.
3. While the section explains the relevance of each metric to the study's goals, further justification on the selection of these specific metrics over other potential evaluation measures could enhance the reader's understanding. For instance, discussing why these metrics were preferred over other common metrics in time-series classification could provide deeper insights into their suitability for evaluating early classification models.
4. The mention of greedy search within the description of the Action Space might indeed be misplaced. It would be more appropriate to discuss in the context of model optimization.
5. The relationship between λ in Eq(5) and λ in Eq(13) needs to be clarified in the paper.

Validity of the findings

1. Given the sensitive nature of educational data and the potential consequences of predictive modeling on student interventions, a more thorough exploration of ethical considerations and potential biases within the model would strengthen the paper.
2. Answer your research question in the conclusions; what did you learn compared with current, significant research.

Additional comments

1. The formatting of the references in your manuscript should be carefully reviewed and adjusted to meet the requirements of the PeerJ journal.
2. The manuscript exhibits instances of informal or inconsistent phrasing in several sections, which could detract from its clarity and professional tone. For instance, Line 281, there is an ε probability. Line 282, with a 1- ε probability. I strongly recommend a comprehensive review of the text to identify and rectify these irregularities.

---

## Round 0.2 · accepted · Accept

· Academic Editor

Accept

The authors have addressed the reviewers' concerns, and I recommend accepting this paper.

Reviewer 2 ·

Basic reporting

My comments have been addressed.

Experimental design

My comments have been addressed.

Validity of the findings

My comments have been addressed.

Additional comments

My comments have been addressed.